# Monitoring the geodynamic behaviour of earthquake using Landsat 8-OLI time series data: case of Gorkha and Imphal

Biswajit Nath <sup>1,2,3</sup>, Zheng Niu <sup>1,2 \*</sup>, Shukla Acharjee <sup>4</sup> and Hailang Qiao <sup>1,2</sup>

<sup>1</sup> The State Key Laboratory of Remote Sensing Science, Institute of Remote Sensing and Digital Earth (RADI), Chinese Academy of Sciences (CAS), Chaoyang District, Beijing-100101, China

<sup>2</sup> University of Chinese Academy of Sciences (UCAS), Beijing-100049, China.

<sup>3</sup> Department of Geograhy and Environmental Studies, University of Chittagong, Chittagong-4331, Bangladesh

<sup>4</sup> Department of Applied Geology, Dibrugarh University, Dibrugarh-786004, Assam, India

Correspondence to: Zheng Niu (niuzheng@radi.ac.cn)

Abstract. Prediction of earthquake in advance is really a challenging task for the scientific community till now. But research results from various scientists regarding lineament extraction using satellite imageries help us to way forward for earthquake monitoring study. For the present study, Landsat 8 OLI Time series data analyzed by integrating four different remote sensing and GIS software's for automatic lineament extraction, its change, including lineament lengths and directions study by creating rose diagrams and finally vertical surface

- transect profile curve drawing. Two recent major earthquakes (in different geological settings Gorkha of Nepal 7.8 M<sub>w</sub> and Imphal, Manipur of Eastern India 6.8 M<sub>w</sub>) epicenter based single tile and corresponding same temporal scenes (three for before and one for after quake respectively) were considered for each case to perform lineament extraction, length variation and vertical surface transect profile change analysis. The research results witnessed major variations in lineament number, lineament length and its trends. The major trends found in an ESE-WNW, N-E, N-S, E-W, NNE-SSW directions and ESE-WSW, ESE-WNW, NE-SW on pre-earthquake scenes compared
- to post earthquake ESE-WNW, NE-SW, NNE-SSW were found for Gorkha, and ESE-WSW for Imphal regions respectively and in both cases, it was observed that the lineation trends return to its earlier status after an earthquake strike. The results obtained using the automated and geo-integrated methods compared cross validation with each other showed our method worked practically for earthquake monitoring and one can apply this new novel combined approach to predict the probable earthquake occurrence in advance just a few days before it strikes.
- Keywords: Monitoring, Geodynamic behaviour, Earthquake, Landsat 8 OLI, Gorkha, Imphal

# **1** Introduction

- In the modern geoscientific time frame, remote-sensing and GIS techniques have been tremendously used for obtaining reliable information from satellite imageries at macro to micro scale level investigations. Studies of linear geologic features (lineaments) from macro to micro level have been increasing rapidly. Lineaments extraction from satellite imagery either by visual or automatic interpretation have been long interest of geologists, where the character and extent of these features have been realized and lineament analysis of remotely sensed data using automatic extraction, is a valuable source of information for studying the structural settings of an
- area. Lineament" the term has been widely used in the field of geology and literally it expresses by different scientists through their

5

research work in different ways. The term lineament was first described as significant line of landscape within the basement rocks (Hobbs, 1911). The lineament defines as linear features in a landscape identified on satellite images and aerial photographs, most likely have a geological origin. Generally, lineaments are underlain by structural zone, fractured zone, a series of fault or fold aligned hills zone of localized weathering and zone of increased permeability, porosity, seismicity, landslide formation (Pradhan et al., 2006), active erosion and karst development (Elmahdy and Mostafa 2012b).

- Lineament extraction and analysis have been studied by different distinguished scientists (Masud and Koike, 2011; Caponera, 1989; Koike, 1995; Mah et al., 1995; Karnieli et al., 1996; Kim et al., 2004; Park et al., 2000, and Costa et al., 2001). Besides these, lineament analysis has been used extensively for geologic interpretation, particularly from the 1930s with the advent of photo geology (Lattman, 1958); because satellite data provides quick and useful baseline information on the parameters controlling the occurrence and movement of
- 10 groundwater like geology, lithology/structural, geomorphology, soils, land use/land cover and lineaments. With the advancement of remote sensing techniques, identifications of lineaments for earthquake have become a rapid and cost effective procedure. One of the main features of geological interpretation of satellite imagery has been the recognition of lineaments varying in length from a few kilometers to hundreds of kilometers (Onyedim and Ocan, 2001). The lineament is a mappable linear or curvilinear feature of a surface whose parts align in a straight or slightly curving relationship (Hung, 2005). The term lineament described as a simple or a composite linear feature of
- a surface whose parts are aligned in a rectilinear or a slightly curvilinear relationship and which differ from the pattern of adjacent features and reflects some sub-surface phenomenon (O'Leary et al., 1976).
   Moreover, (Casas et al., 2000) lineament mapping and analyses have been gaining popularity with the increasing availability of satellite
- images. Since satellite images are obtained from varying wavelength intervals of the electromagnetic spectrum, they are considered to be a better tool to discriminate the lineaments and to produce better information rather than conventional aerial photographs. Lineaments are simply linear or curvilinear edges that may be related to geological structures (faults, joints, line weakness), geomorphological features (cliffs, terraces, linear valleys), Tonal contrast due to (vegetation, soil moisture, rock composition) and human activities and/or constructions (roads, tracks, building, mining). In this particular study, we are trying to get information regarding the probable earthquake occurrence by extracting and analyzing geological features i.e. lineaments, lineament length change and vertical transect profile extraction from Landsat 8 OLI satellite imageries.
- As we had recently faced two major earthquakes one from Gorkha, Nepal and another Manipur, Eastern India, so, we decided to cross examine its feasibility and to test the method and approaches which would be a quick method to know the geodynamic behaviour of earthquake occurrence and it could be successfully applied in any regions of the world to test the probable earthquake occurrence in advance. The main purpose of this study was to extract lineament features through automatic approaches and to know the changes of lineament status based on before and after earthquake satellite scenes of Landsat 8 OLI Data which struck in Gorkha, Nepal (7.8 M<sub>w</sub>) and
- 30 Imphal, Manipur (6.8 M<sub>w</sub>) regions of India in the year 2015 and 2016 respectively, and to overlay this lineament result with the epicenter of the two major earthquakes and along with it, to figure out surface change considering the same scenes by drawing arbitrarily vertical surface transect profile over the study areas and considering these geodynamic behaviour as valid probable earthquake occurrence signals by interpreting series of Landsat satellite imageries before an earthquake strike, and this test has been done using Landsat 8 OLI time

series satellite data to justify whether we can predict or not of the probable earthquake occurrence in any area of the world and also to find out the situation of lineament and its length change of pre-and post-earthquake.

The main objective of this study is to find out the underlain geological change due to the earthquake by considering lineament change detection and directions analysis using the consequences of pre (3 images) and post-earthquake (1 image) Landsat 8 OLI time series satellite imageries and to predict the probable earthquake occurrence in advance by analyzing consecutive satellite images prior to strike.

2 Data and Method

#### 2.1 Study Area and Data

For our present research, initially study site was selected in the Gorkha, Nepal earthquake region and for our method validation later we had considered and tested for another earthquake which occurred in Imphal, Manipur region to know the lineament change of that area and

10 to justify its significance what lineament change actually happened prior to an earthquake strike and the resulting phenomena change in the post-earthquake time. For this study, we have chosen 4 satellite scenes of the same area for each individual case based on the epicenter. The study area covers 370 km<sup>2</sup> in size of each single satellite scene.

The Epicenter (earthquake epicenter information accessed through recent earthquake databases from USGS) based single tile image (three for before and one for after earthquake change monitoring) downloaded from the USGS Landsat archive to see the changes of the

15 lineaments and to get an idea whether this lineament change and vertical transect profile would be an ideal choice or not to predict a probable earthquake occurrence in an area.

The first study case is Nepal, which lies towards the southern limit of the diffuse collisional boundary where the Indian plates under thrusts the Eurasian plate, occupying the central sector of the Himalayan arc (Jonathan, 2015). The lasting time of 50 seconds, Nepal earthquake occurred on 25 April, 2015 at a depth of 15 km with a magnitude of 7.8 which referred as shallow earthquake and created massive

- destruction. And its epicenter was 28.147° N and 84.708° E. According to the (USGS Earthquake Hazard Program, April, 2015) the earthquake was caused by a sudden thrust, or release of built up stress, along the major fault line. And the second case study is Imphal, Earthquake which is part of northeast regions of India in the state of Manipur on January 4 with a magnitude of 6.7 and a maximum Mercalli intensity of VII Category: Very Strong:, and the location of the epicenter was 24.804° N and 93.651° E (2016 Imphal Earthquake-Wikipedia, 2016) and as a resultant of it, areas affected were Bangladesh, India, Myanmar, Nepal,
- Bhutan and at least eleven people were killed, 200 others were injured and numerous buildings were damaged. The study areas and the data sources are shown in Figure 1. The Landsat 8 OLI radiance corrected imageries of pre-and post-earthquake for both regions shown in Figure 1A and 1B, which was downloaded through USGS Landsat archives and later pre-processed with sensor information from metadata file using the ENVI 5.3 software.

# 2.2 Overview of the Combined Method

In this paper, the automatic lineament delineation was based on a decision of the most appropriate band for edge enhancement, followed by an edge sharpening enhancement technique which gives the best result of lineaments that are not delineated by human eyes and apply LINE module of PCI Geomatica 9.1 v to recognize lineaments. Landsat 8 OLI satellite data were used considering the best band for

lineament extraction. The lineament extraction algorithm of PCI Geomatica software consists of automatic (or digital) extraction process using various computer aided methods for edge detection, thresholding and curve extraction steps (Abdullah, et al., 2010). These steps were carried out over derived principal component analysis image i.e. PC1 of the study area under the default parameters windows where user defined modification of values can be done with this software.

5 Consequence map generated for Nepal and Manipur Earthquake using 8 imageries where 4 images considered for each area (3 successive satellite scenes for the before earthquake and 1 for after earthquake evaluation). Before the methodological description breakdown, the overall workflow of the present research is highlighted in Figure 2.

# 2.2.1 Preprocessing and PCA performed using Landsat 8 OLI Imageries

Satellite images (multispectral or digital elevation models) and aerial photography are broadly used to extract lineaments for different 10 purposes, like defining geological structures and tectonics fabrics. The first principal component image (PC1) of Landsat 8 reflected bands

is a good example; as PC1 carry most information and is suitable for lineament extraction purposes. Lineament distribution in Manipur and Nepal (earthquake epicenter and its surrounding areas) were prepared using Landsat 8 OLI satellite imageries. Images consist of 11 spectral bands with all sizes (30\*30 m). The OLI spectral band in gray scale was used. The scene size is 185 km north-south by 185 km east-west. The date back to 20th March, 5th April, 21st April and 7th May of the year 2015 for Gorkha,

Nepal and 30th November, 16th December of 2015 and 1st and 17th January of the year 2016 in Imphal, Manipur are considered for this present research.

A lineament map is extracted from Landsat 8 OLI satellite imagery using different integrated techniques of Remote sensing and GIS through four different software; where one output is used by other software to get out the final lineament results. In this experiment, automatic extraction of geologic lineament performed through different steps starting from raw satellite imagery preprocessing like DN

value conversion into radiance, next radiance to reflectance and later FLAASH Atmospheric correction and Principal Component Analysis (PCA-1) using 1st band was performed using ENVI 5.3 software (Figure 3).

# 2.2 Extraction of Lineaments

There are two common methods for the extraction of lineaments from satellite images i.e., visual and automatic extraction. For visual extraction, the user first starts with some image processing techniques to make edge enhancements, using the directional and nondirectional filters such as Laplacian and Sobel, then the lineaments are digitized manually by the user and for automatic extraction various computer-aided methods for lineament extraction had been proposed and used in different research.

Most methods are based on edge filtering techniques. The most widely used software for the automatic lineament extraction is carried out with LINE Module of PCI Geomatica (Kocal et al., 2004). In this study, PCI Geomatica 9.1v software has been deployed for digital

analysis and automatic extraction process. The algorithm consists of 3 stages (PCI Geomatica Training Manual, 2015). A lineament distribution over the study area extracted using the LINE command technique and lineament extraction algorithm of PCI Geomatica software consists of edge detection, thresholding and curve extraction steps. The algorithm parameters and its corresponding values which are used for processing are mentioned below: RADI-Radius of the filter in pixels (10), GTHR-Threshold for edge gradient (50), LTHR-

Threshold for Curve length, in pixels (30), FTHR-Threshold for Line fitting error in pixels (3), ATHR-Threshold for Angular difference in degrees (30), DTHR-Threshold for linking distance in pixels (20).

The main geometric characteristics of a single linear line are orientation and length (continuity) and in case of curved line, curvature (Jordan and Csillag, 2003). This program reads line endpoint data and generates a directional diagram that depicts the orientations of the

5 linear features. Later, the Splitting command line at their vertices and lineament line length is in meters which is then converted in the kilometer level using ArcGIS 10.2.2 software. The most important factor for this was that the lineaments in an automated one were shorter in length so that a few of them could be combined to form one long lineament.

#### 2.2.3 The line split generation using model builder

The renowned GIS developer ESRI developed Model builder which is used to automate GIS processes by linking data input, ArcGIS Tools/functions, and data output. Model builder is part of the ArcGIS geoprocessing framework. The main advantage of using the model builder for GIS work is that extracting processes can be automated without using any code and another advantage is that it can save GIS processes and rerun the model at any time, whenever it's required. The final step is running the model and view the data output (Tutorials ESRI-ArcGIS 10.2.2-Model Builder). Model Builder is an application which used to create, edit and manage models. Models are workflows that string together sequences of geoprocessing tools, feeding the output of one tool into another tool as input. Model Builder can also be thought of as a visual programming language for building workflows.

- However, a model is nothing more than a sequence of tools and data chained together. When a model has been saved, it becomes a model tools. Model Builder open by clicking geoprocessing multiple way data can be added to the model builder canvas by dragging or clicking on add button. From the ArcGIS toolbox, the split line command is used to break the line which drags into this model, thereafter, applying add connection tool for model execution. Once connected to the tools can execute the model from within the model builder finally by
- 20 clicking Run button. Next model is saved with the fruitful name and then is ready for next level operation (Figure 4). This lineament features extracted as a compound line split into a single line at their vertices and after that, this split lineament features are converted into two formats i.e., a CAD dxf format which is used by RockWorks 16 software to perform lineation computations for Rose diagram preparation to get the directions of the lineaments and Shapefile format for a number of lineament, and its length analysis.

## 2.2.4 Lineament length analysis method using ArcGIS

25 In this stage, we created a field to get lineament length of all the attribute values using calculate geometry under corresponding attribute table and used projection information automatically from the data sources and unit considered in Kilometer. After getting lineament length, quantities choose for symbology generation where the quantile method was applied as for its classification using graduated color ramp to know the length variation (lowest to highest length) over different temporal scenes over these regions and in this way, same method was applied on the remaining temporal scenes over the two study areas.

#### 2.2.5 Lineament directions analysis method for creating rose diagram

RockWorks 16v software come completely with a rich array of Rose diagram and lineation analysis utilities. It explains the frequency of lineation in a given orientation. For trend, directional analysis we have used RockWorks 16v software where previously saved lineament data as dxf format used as an important file under the Linear Tab options, then processed and changed linear units in UTM Meter, after

- that assigned projections parameters and later by using this data file, lineament computation has been performed to measure Bearing (unidirectional: 0-360 degrees), Length (Meter) and/or midpoint. X1, Y1 select the names of the columns in the data sheet that contain the X and Y coordinates for the beginning point of the lineation. This can be Easting in meters or feet, decimal, longitude, etc. and X2, Y2 select the columns that contain the X and Y coordinates for the endpoints of the linea and at the final stage Rose diagram has been prepared based on Endpoint data.
- The directional diagram that depicts the orientation of the linear features and saved it in the required format as a tiff file. By following this method, we have generated all the remaining rose diagrams for these two earthquakes to figure out the directional change of lineaments due to earthquake.

## 2.2.6 Vertical transect profile drawing method

In this section, we have drawn vertical transect profile on each satellite scene following arbitrarily profile section using ENVI 5.3 surface profile extraction tools based on the false color (5, 4, 2) combination where 3 transects (vertical) were drawn selecting surrounding epicenter i.e. left side (free to select), central is fixed over the scene and right (free to select). Both left and right section transect move based on Epicenter of the earthquake. These vertical sections were drawn to support lineament change study and to observe surface movement before an earthquake strikes and also to check post-earthquake scenario.

# 2.2.7 Overlay and buffer analysis method

Finally, we have created an overlay analysis of four temporal time frame shapefiles in an integrated manner applied for both case studies to observe the overall change and after that, specifically to observe epicenter and non-epicenter based change of lineaments by considering immediate pre-and post-quake lineament data. In this analysis, we have generated 100 km buffer and outside 100 km buffer zone map where, based on the buffer distance, clip was performed on individual layer in both cases and finally overlay operation was observed to know lineament change of pre-and post-earthquake using ArcGIS 10.2.2 software.

#### 25 3 Results

Before proceeding, here we declare that; northern part of this satellite scene is covered by parts of China. On the other hand, Imphal earthquake's epicenter was in Manipur's Tamenglong district, bordering area with Myanmar are in the right section of that corresponding image with depth measured 55.0 km and this Imphal earthquake occurred as the result of strike slip fault in the complex plate boundary region between India and Eurasia plate in Southeast Asia. In the region of the earthquake, the Indian plate is moving towards the north-

30 northeast with respect to Eurasia at a velocity of approximately 44 mm/yr. The regional plate boundary in eastern India-the Indo Burmese Arc is oriented approximately south-southwest-north-northeast (USGS-Tectonic Summary, 2016).

The following different sections represents the present research derived results based on pre and post-earthquake scenarios where firstly, it highlights automatic lineaments extracted data along with lineament length change and statistical information; secondly, Rose diagram is opted to check the directional changes over the study regions; thirdly, vertical transect profile extraction based on satellite scenes and

5 finally, overlay analysis of lineament layer to represents before earthquake scenarios compare to post earthquake lineament change and also tried to analyze with earthquake epicenter which helps to make decision regarding earthquake occurrence in advance.

## 3.1 Lineaments, Line Length and Statistical Analysis

- The focused study areas are tectonically active in nature and recently vibrating the earth's surface with high to moderate magnitude earthquakes which altered the geological settings in the areas as we noticed from different research. As no research, has been done considering lineament change and vertical transect profile extraction approaches we tried to test it using recent dataset. To quantitatively evaluate our proposed combined method, lineaments are automatically extracted using the image after Principal Component Analysis (PCA).
- The resultant lineament map and its line length converted in kilometer later it quantiles in 5 class range to know the length variation conducted for all temporal imageries (8 scenes in two target case areas) along with corresponding statistical information's are shown in Figure 5 (Nepal), Figure 7 (Manipur) and Table 1 and Table 2 respectively. All of these are illustrated below in two segments; i.e., Pre-and Post-earthquake scenarios.

#### 3.1.1 Pre-earthquake Scenario

The above indicative results of lineament analysis used in Landsat 8 OLI temporal data illustrates the scenario of Gorkha, Nepal earthquake shown in the Figure 5 and Table 1. Our extracted data and three Maps prior to the earthquake strike suggests that, a major number of lineament exists and it varies prior to earthquake hit in the study area. As the earthquake hit on 25th April 2015, our focus was to find out the situation of lineament where, 36 days and 20 days' prior earthquake the number of lineaments was found decreasing from

25 initial lineament and just 4 days before it tremendously increased and after striking earthquake, the number of lineament decreased compared to its preceding values. After checking lineament numbers we checked the lineament length which is converted in kilometer where we found quite similar scenario as observed in case of lineament number, except 4 days prior data where maximum length was dropped suddenly (Figure 6).

Whereas, same method applied over Imphal, Manipur Earthquake (6.8 M<sub>w</sub>) to know the lineament change of pre and post earthquake.
 Actually, this earthquake site was chosen for cross validation study with Nepal, whether, our combined method will work or not over time and space.

The results of the Manipur earthquake depicts that, number of lineaments followed the same trend decreasing-increasing (20 days of prior to the earthquake) in the first initial image (36 days of pre quake) and drastically increased prior to 4 days of the earthquake and sum value

of all lineament length also followed a similar trend decreasing-increasing and increasing scenario after earthquake (Figure 7 and Table 2).

In the following Figure 8 shows the descriptive staistics based bar diagram of lineament frequency of pre earthquake event of Manipur region.

## 3.1.2 Post-Earthquake Scenarios

In the Gorkha, Nepal case, lineament data of after strikes suggests that, it was readjusted naturally and found a void of lineaments where

the epicenter is located even which they exist in an immediate pre earthquake scene on 21 April, 2015 and the total lineations of length was dropped drastically from 21973.44 to 16382.09 kilometers after an earthquake, assessing over the whole scene (Table 1 and Figure 9-9a).

Whereas, an exception is observed in an aftermath situation of lineaments in Manipur case. The lineament and length variation statistics suggest increment of lineament and it increased near about double from its initial data (Table 2 and Figure 9-9b). In the following Figure

(10-a and b) shows the descriptive staistics based bar diagram of lineament frequency of post earthquake event of Nepal and Manipur regions respectively.

## 3.2 Rose Diagram for Directional Change Measurement (based on Overall Scene)

In this work, our study area was associated with a rose diagram of directions of lineaments. The interpretation of all lineaments based on before and after earthquake data generated for the two earthquakes regions which was badly hit in those areas with different magnitudes.
For our research, we are trying to observe the lineament direction change based on pre and post earthquake scenes and to cross validate each other whether this change data can be a great indication or not prior to earthquake strike. The case wise interpretation results in a directional change of lineaments are shown in Figure 11 (11a for Gorkha, Nepal and 11b for Imphal, Manipur).

#### 3.2.1 Pre-Earthquake Scenario

# a. Gorkha:

- According to rose diagram of this earthquake, initial data of 20 March (36 days before) showed lineament directions were in an ESE-WNW, E-W, NNE-SSW and N-S positions and 5 April data (20 days before) tells two major trends which direction was ESE-WNW and NNE-SSW and another one N-E may also have a role in it and directional change firstly noticed on that image. Whereas, on 21 April prior to Earthquake (4 days before), two major trends clearly be interpreted which are ESE-WNW and N-S. Subsequently, all these lineament directional trends were correlated and related within the regional context of this area which is a great indication of any structural change
- and can be a vital clue to know that earthquake is approaching very soon in that region. (Figure 11-11a).

# b. Imphal:

On the other hand, the major lineament data trends from Imphal, Manipur regions on 30 November, 2015 (36 days before) suggests, ESE-WSW direction and on 16 December 2015 (20 days before), it showed a major trend to be ESE-WNW along with considering bin lengths another trend of NE-SW can also be exists. Besides those, on 1 January 2016 prior to 4 days of earthquake events, two major trends NE—

30 SW and ESE-WSW were identified by interpreting the lineament data (Figure 11-b).

5

20

#### 3.2.2 Post-Earthquake Scenario

The extracted lineament data of the post earthquake period for two regions have clearly indicated a directional change of lineament. In the first case of Gorkha, where, the major lineament directions changed after the earthquake was ESE-WNW direction along with two other trends were NE-SW and NNE-SSW indicates the movement of the earth's surface (Figure 12-12a). On the contrary, for Manipur, the major trend of lineations directed to ESE-WSW position. (Figure 12-12b).

## 3.3 Vertical Surface Transect Profile results based on Landsat 8 OLI Time Series Data

Vertical Surface Transect Profile draws performed in three different parts of each image, including along the Epicenter of the earthquake on all time series Landsat 8 OLI Radiance Corrected Images. These vertical transect profiles were automatically extracted using arbitral

transect drawing using ENVI profile extraction tool bar, where, same assessment approach was followed to know the changes of surface prior to the earthquake and compare to post earthquake profile.

Later on, we had focused on to extract vertical surface transect profile data (top to bottom approach) for our two study areas. These three transect sections were named as Left Vertical Transect (LVT), Middle Vertical Transect (MVT) and Right Vertical Transect (RVT) sections (Figure 13 and 14) where X-axis data represents vertical surface transect distance value (in meter) and the Y-axis value represents

pixel level fluctuations. However, these sections gave us a vital clue of these two earthquake events where we had also noticed the abnormality of profile sections before the pre earthquake event in both the cases.

#### 3.3.1 Pre-Earthquake Scenarios

In this Gorkha (Nepal) case, we had noticed that, in pre-earthquake scenes 20 days before the surface had been starting to vibrate and compressed and spiky jumps found in naturally in all three sections which differ than first image and just 4 days prior to earthquake strike, surface strain was extended (Figure 13).

On the contrary, Imphal (Manipur) case, we had also noticed an abnormality throughout the whole scene where in LVT section, the epicenter was formed along that line and sudden spiky jumps and abnormality of surface signature had been identical, later the surface strain starts to move throughout the epicenter region and compressed and spiky jumps found in MVT section where cloud presence was

25 observed even in an initial image surrounding the epicenter regions. This abnormality of data had been checked later with post earthquake scenes which is shown in Figure 15-15b.

#### 3.3.2 Post-Earthquake Scenarios

In this section, in the Nepal case, profile abnormality is observed in the RVT section, where it is found in X-axis within the range of 2500-30 4500 meter and in Manipur case, post-earthquake profile showed, that the earth has its natural tendency to readjust and try to regain its previous condition, but its abnormality still exists in the MVT section of the scene where urban area is located and it is found within the 2000-6000-meter distance range.

# 4. Discussion

## 4.1 Comparison of two earthquakes considering Lineaments and Its Length Change

The automatic extracted lineament data values of both tables varied, which is prepared based on temporal dates images maintaining similar days' interval for two earthquake cases while image selection and the derived result suggests, two different regions had produced different number of lineaments as observed in overlay techniques using ArcGIS 10.2.2 (Figure 16: a-Nepal; b-Manipur) and line length value differ (Figure 6 and 7) due to different geologic condition, structural arrangements, depth and magnitude variations.

## 4.2 Buffer Analysis based on Epicenter (100 km) and Non-epicenter (outside 100 km)

In this section, we have focused on overall lineament changes based on buffer distance measured from epicenter 100 km and outside 100 km based buffer for two earthquake regions shown in Figure 17 (Nepal) and Figure 18 (Manipur) respectively.

A clustered bar chart of Gorkha (Nepal) earthquake showed (Figure 17-17c) immediate pre and post earthquake lineament change measured from corresponding epicenter, where, bar chart represent that, before earthquake very few lineament exist within 100 km buffer distance compared to post quake, whereas, in outside 100 km buffer zone, lineament was found 20,263 in number in immediate pre

- earthquake time and later reduced in number 17,844 lineaments compared to 100 km buffer zone (7992 in number of lineaments). Whereas, in Manipur case, number of lineaments increases in the 100 km buffer zone in the post earthquake time rather than 4 days prior to earthquake strike (Figure 17-17d and e). However, in the outside 100 km buffer zone, lineaments were more (28985.04 in number) rather than pre quake time (Figure 17-17f). It is worth mentioning that, this post quake outside 100 km buffer zone lineaments are even
- more higher than within 100 km buffer zone of post quake time.

# 4.3. Vertical Surface Transect based Analysis

Due to cloud presence over epicenter and its adjoining area, the profile showed major and abrupt changes within the vertical distance range of approximately 4000-6000 meter for Nepal (Figure 13). This abnormality of data has been checked later with the post earthquake scene

- as shown in Figure 15-15a. Finally, we have been fcoused on Manipur where, 4 days before surface profile data showed high compression and close to each other along the line of epicenter (LVT section, Figure 14-b) rather than 20 days before where high abnormality exist due to cloud appearance over that regions and it found in approximately 3000-6000 meter distance. Besides this, in the MVT and RVT sections of all pre quake profiles clearly showed surface anomalies with few exceptions of high profile peak generated due to cloud appearance. Later these data also compare with post quake vertical profile where it was found in normal condition like in 36 days before scenario
- (Figure 15-15b).

From the overall observations of the present research (based on the results coming out in each section), we have found lineament change and vertical transect surface profile anomalies over both the study areas which were clearly discernable prior to 4-20 days before earthquake. So, the overall integrated analysis suggests that, if we use all the data in time then surely we can tell about probable earthquake occurrence in advance prior to its strike.