# Peer review of "Monitoring the geodynamic behaviour of earthquake using Landsat 8-OLI time series data: case of Gorkha and Imphal"

_Natural Hazards and Earth System Sciences, 2017_

## Referee Comment (RC1) · Anonymous Referee #1 · 15 May 2017

Authors face a quite important (even if not completely new) scientific issue in a frontier research field, i.e. the use of multi-temporal satellite images analysis for investigating lineaments changes in a possible relation with impending earthquakes. Despite several elements of interest the paper suffers of several elements of weakness that make it not acceptable in its present form. 1. Authors consider the "anomalous" change in lineaments length&distribution without whatever reference to a "normal" behaviour and whatever test in seismically unperturbed conditions devoted to characterize the considered indicators (and measuring procedures) in terms of expected values and normal variability. As just few image before and after earthquakes are considered no information we have to evaluate the stability of the measured indicators (and procedures) in

absence of seismic event and to compare the intensity of observed fluctuations to the ones normally observable as a consequence (for instance) of image-to-image changes for observational conditions (atmospheric correction models, especially if performed in absence of appropriate information on local atmospheric conditions, can also amplify instead that reduce such variability ). 2. For the same reason (no attempts to verify if similar "anomalies" are actually absent in absence of earthquakes) results actually achieved in just 2 cases do not support firm statements like the one given in the abstract: "The results obtained using the automated and geo-integrated methods compared cross validation with each other showed our method worked practically for earthquake monitoring and one can apply this new novel combined approach to predict the probable earthquake occurrence in advance just a few days before it strikes". 3. Quality of figures is very poor (very often not supported by legends explaining their content as well as the use of colors, always with numbers too small to be readable) and their full understanding not always possible. 4. Important points of the analysis are not explained at all (for instance how authors manage the evident presence of clouds and snows in the images and how they avoid their variable presence affects also lineaments variability estimates). 5. English is generally very poor specially in the use of verbs and a close review by an English mother tongue reader is required to make the text not just grammatically correct but, at least, understandable in several points.

---

## Author Comment (AC1) · 16 May 2017

Dear respected Anonymous Referee #1

Greetings and Happy to receive your valuable interactive comments over examination of our submitted manuscript on "Monitoring the geodynamic behaviour of earthquake using Landsat 8-OLI time series data: case of Gorkha and Imphal" by Biswajit Nath et al. which Received and published on: 15 May 2017. I am submitting my initial reply (on behalf of all authors) against all of your valuable comments.

As this manuscript need further improvement, so, we need few weeks to complete all new works for revised submission (i.e. considering one New image for Normal variability status proof of both cases to represent its stability compare to present considered images) with better visualization of maps with corresponding legend, therefore we hope it will be readable and understandable by scientific community. However,

Comments 1 by Anonymous Referee#1:

Authors consider the "anomalous" change in lineaments length&distribution without whatever reference to a "normal" behaviour and whatever test in seismically unperturbed conditions devoted to characterize the considered indicators (and measuring procedures) in terms of expected values and normal variability. As just few image before and after earthquakes are considered no information we have to evaluate the stability of the measured indicators (and procedures) in absence of seismic event and to compare the intensity of observed fluctuations to the ones normally observable as a consequence (for instance) of image-to-image changes for observational conditions (atmospheric correction models, especially if performed in absence of appropriate information on local atmospheric conditions, can also amplify instead that reduce such variability ).

Reply against Comments 1 by Authors: Based on your comment, we will consider one more image before (for each case) the earthquake to show the nornal behaviour of lineament in the absence of seismic event along with our existing anomalous condition with intensity observed fluctuations in normal situation and abnormal situation (comparing image) of before and after earthquake what we observed prior to both the earthquakes. Before extraction of lineament we had performed FLAASH atmospheric correction using ENVI 5.3 software. However As the satellite images suffers few cloud, and we think it not disturbed and reduce our lineament data too much as the automatic extraction process generate sufficient number of lineament features. However, we will clarify and explain it further after adding New image results for both cases where we will show the normal status of it and accordingly we will improve the text.

Comments-2 by An. Referee #1:

For the same reason (no attempts to verify if similar "anomalies" are actually absent in absence of earthquakes) results actually achieved in just 2 cases do not support firm statements like the one given in the abstract: "The results obtained using the automated and geo-integrated methods compared cross validation with each other showed our method worked practically for earthquake monitoring and one can apply this new novel combined approach to predict the probable earthquake occurrence in advance just a few days before it strikes"

reply against comments-2: In our first reply,accordingly we will consider new image where output will represent the absence of such anomalies and thereafter, we hope the output and present existing results both will finally able to highlight and support the statement which we mentioned in the abstract or subject to further modification of abstract according to output results.

Comments 3 by Ann. referee#1:

Quality of figures is very poor (very often not supported by legends explaining their content as well as the use of colors, always with numbers too small to be readable) and their full understanding not always possible.

Reply against Comments 3: We had saved the all images in Tiff 300 dpi but it changed probably while multiple copying of images. However, we will improve the legends font size and improve image color quality also and we hope it can be visualize and readable.

Comments-4 by Anonymous Refer.#1

Important points of the analysis are not explained at all (for instance how authors manage the evident presence of clouds and snows in the images and how they avoid their variable presence affects also lineaments variability estimates).

Reply against Comments-4:

We will follow the suggestions and important points of the analysis will be explained along with evident presence of clouds partially over the images. Presence of cloud is a

fact, prior to earthquake strike, so we will clarify it in our revised work submission file.

Comments-5 by Anony. Refer#1

English is generally very poor specially in the use of verbs and a close review by an English mother tongue reader is required to make the text not just grammatically correct but, at least, understandable in several points

Reply against comments-5: We will follow your suggestion and accordingly improve the text of whole manuscript and revised our submission after finishing the new image task to show the normal behaviour while it will shows the absence of earthquake scenario.

So, finally we need time for our revised work and then you will find our improvement manuscript according to your fruitful comments.

Thanks once again for your valuable comments.

Thanking You Sincerely yours Biswajit Nath (on behalf of all authors)

---

## Referee Comment (RC2) · Anonymous Referee #2 · 19 May 2017

**Review of the manuscript: "Monitoring the geodynamic behaviour of earthquake using Landsat 8-OLI time series data: case of Gorkha and Imphal", by Biswajit Nath et al.**

The paper investigates the possible relationship of lineaments (number, length, main orientation) and their changes, extracted from satellite imagery, with geodynamic behavior in seismically active zones, in particular studying scenarios before and after important earthquakes occurred in Nepal and Eastern India.

Authors used commercial software and standard routines and tools to perform imagery analysis claiming their "method worked practically for earthquake monitor and one can apply this new novel combined approach to **predict** the probable earthquake occurrence in advance just a few days before it strikes". The topic is surely of interest but I don't agree with authors' conclusions which, from my point of view, are not confirmed by the presented results nor supported by a methodological approach with sufficient scientific soundness, according to my following comments and remarks. For these reasons, I cannot recommend publication of the paper in its present form.

Main comments:

Generally the English is very poor through all the paper. Many sentences are very long (see for example page 2, lines 28-35(!), but there are many others) and often written in an awful language, making them hardly readable and understandable.

1. Methodology

   Authors generally use commercial softwares and standard routines and tools, often providing description and details (e.g. they spent quite a paragraph to explain what is the "model builder" of ArcGis) which are not very new nor useful. On the other hand, they do not provide details about the theoretical background they have in mind to investigate the geodynamic behavior of the study areas. Why they are studying number, lengths and direction of lineaments? Why do they expect variations in these features (and to what "sign" and extent) and what are the theoretical models explaining their relationship with Earthquakes? Why do they investigate "vertical transect profiles"? What do they exactly means with "vertical profiles" and why they would expect variations in "false color (5, 4, 3)" imagery? Again, what is the theoretical background behind this analysis? If models exist, authors should strictly refer to these and verify if the model hypotheses are confirmed or not by their experimental results. On the other hand, if models do not exist yet, authors should first present their theoretical model and then demonstrate their results are coherent with or not.

   Moreover, the scientific soundness of the presented methodological approach, based only on a few images (3 before and 1 after the quake) is quite poor. How these (very limited) conditions, may drive authors to so firm conclusions? A more extended dataset should be analyzed and it is absolutely required in my opinion to provide more convincing and scientifically rigorous results. For instance, no confutation analysis (lineaments variations in absence of EQ) is provided. To do that authors should analyze similar temporal sequences of

LANDSAT imagery in periods when no EQs occurred in the study area, showing that similar "changes" did not occur in these circumstances.

2. Result Presentation

The quality of images is very poor and the way they are commented in the text and within the captions is often not very helpful for the reader.

3. Result interpretation

Authors are interpreting their results only in terms of consequences/effects of EQ. No mention to possible artefacts or unwanted effects due for instance to changes in observation conditions (e.g. clouds, snow, atmospheric or surface changes) is done. In my opinion, the observation/illumination conditions may have a significant impact on feature extraction, as clearly demonstrated by the results shown in Figure 5: the lineaments map of April 5 (central map) is very different from the 20 March (left) and 21 April (right) ones. In particular, looking at the bottom of the 5 April map, no lineaments are extracted for this day, whereas a number of features is clearly visible a few days before and after. Do really the authors may assert that such changes are TOTALLY due to geodynamic causes only? May they categorically exclude any other cause (including simple observation conditions)? Moreover, how much the errors and inaccuracies of the processing steps (e.g. lineaments extraction, segmentation and length estimation) impact on the achieved results? Error impacts and algorithm inaccuracies are not discussed at all. These are just some examples of how authors seem to get conclusions and interpretations not really in line with the actually achieved results.

Other examples are in results reported in:

- Table 1: authors achieve different behavior of lineament changes (decreasing-increasing and then decreasing after the EQ for Nepal, but decreasing-increasing and then increasing again after EQ for India) but they still interpret these data as showing "similar trend".

- Figures 13 and 15: where do authors find "anomalies" on profiles? Why should they occur? What do they mean and how do they can be related to geodynamic processes?

- Figure 17: again, a different behavior in the buffer analysis for the two test cases: how do authors may explain this?

---

## Author Comment (AC2) · 20 May 2017

Dear Respected Anonymous Referee#2 Greetings and Happy to receive your valuable in depth comments regarding our paper. According to your provided comments we will start to rectify the manuscript, based on your suggestions, we are now considering another one image (one for each case) to represent there is no anomaly exist in absence of earthquake, while the anomaly was identified by the successive scenes before and after earthquake. We are planning to show the normal behaviour of lineament, compare to abnormal behaviour. However, it will clearly highlight the lineament change over the study areas when we overlay of lineament feature.

As the manuscript need improvement of English, so we can assure you, we will improve

the text and change the maps accordingly wherever it required, So that, it can readable and understandable by the readers.

1. reply on Methodology: Yes, we used multiple commercial software to generate this data from multi-temporal Landsat scenes (i.e., from Preprocessing to final outcome). We will rectify the text regarding the Model builder explanation as it not required based on comments.

As we had observed that, most of the lineaments extraction relevant papers from numerous authors were developed purely based on ground water exploration and geological investigation in different parts of the world and very few papers found regarding lineament interaction with earthquake and no past research work found in our two case study areas.

However, we didn't develop any theory before conducting this research, if required we will try to develop a theoretical logic behind this present work according to our data extraction from successive satellite scenes of Landsat 8 OLI imageries. We are studying the lineament to observe the earthquake, with our question in mind, that, can lineament change observation through successive images helpful for earthquake study, and if it tell the abnormal behaviour, (i.e., whether the total number is increasing, decreasing or stable prior and after earthquake.) and if the total number changes then we adopt the logic that, can length vary and direction of lineament change as the earthquake progresses and advance to strike in the particular area, and if variation found and movement can be identified through the directional diagram( i.e, rose digram we used for it), along with vertical surface profile we drawn in three different parts of the image (based on false color 5, 4, 3 band combination to get the profile change in different bands) as we plotted one in the epicentre, and two others on left and right side of epicentre, (to observe if any surface change exist in those images along the epicentre and its surrounding zones) for that, we had constructed the vertical section profiles over these image. We will clarify it in the text and accordingly interpret the profiles why we considered this signature for earthquake investigation.

There is no established model we follow for our research, its our own think tank to represents the data and try to establish our method based on the outcome we got from the generated figures and its corresponding statistics., However, we feel, we will improve all the figures and corresponding text to catch the general readers for easy understanding.

We will briefly clarify our aim why we investigate these two earthquake which already stricken in those areas, to know geodynamic behaviour of the earthquake through this lineament change observation.

We will compare with the another image which we believe is enough to find out the difference between normal sequence when no earthquake was observed in those areas, compare to abnormal condition which we observed, however, we will prepare and demonstrate the maps according to compare with each other, thereafter it can be visualize through maps and for scientific soundness we will take care of it through out the manuscript.

Cloud and snow presence was not highly deteriorate our lineament data, though it exist in the images, its a common phenomena during the earthquake as we investigated through the image for these two cases. So we can't masking the cloud and snow, and automatic lineament extraction process using PCI Geomatica Line algorithm method have enough potentiality to extract the data, however, we will clarify it in our revised manuscript as the work need to add new image result and comparison based on your suggestions. First we visually observe and then decide to extract the data and created the consequences result from lineament (i.e, lineament length, directions) and vertical transect profile for surface change signature which also indicate the abnormal behaviour prior to the earthquake compare to post earthquake scenario and after getting our new image result, we will according revised our manuscript and clarify wherever it necessary. and we will discuss further the lineament extraction processing steps as a whole which we performed by considering multiple softwares platforms where one image output was considered by another software (Remote sensing and GIS integration)

We will finally modify and improve our text and figures including interpretation results will be shown in sequential way for better understanding.

We will clarify through the normal and abnormal behaviour of lineaments in the presence of earthquake or in absence and also through the profile where circle will be used to mark the abnormality of the and same we will apply on lineament maps to represents the normal and abnormal sequences.

Later, at the end we checked with buffer analysis based on epicenter and non-epicentre of the image to check the lineament change in closely around the epicentre through the successive imageries and can the anomaly exit outside 100 km buffer zone to see the particular change of lineament during the earthquake phase.

However, we will clarify all the above mentioned issues which we observed through your valuable comments and need few weeks to improve further our manuscript and thereafter we will submit to you the revised version where you will surely find our clarification what improvement done on the manuscript.

Once again, on behalf of all authors, I am giving you especial thanks for your close observations on our manuscript and happy to receive your nice comments for further improvement of this time oriented research.

Thanking You

Sincerely Yours Biswajit Nath (Author)

---

## Author Comment (AC3) · 20 May 2017

Dear Respected Sir, Greetings

To The Editor, As we have received two of the anonymous referee's comments, where we found some valuable observations from two anonymous reviewer's end and accordingly we send our initial reply. Both suggested to consider new image to proof where there is no anomaly in absence of earthquake compare to earthquake presence where we observed the anomaly.

However, as they told to improve the manuscript , and we are accordingly start to work on the comments and will revised the manuscript based on their suggestions.

[Figure]

So therefore, we need time from your end for further improvement works and according revised submission to them.

Looking Forward to hear from you regarding the time which we badly need for improvement of our manuscript so, that, finally it will be more scientific and sound while we clarify the comments.

Thanking You, Sincerely yours Biswajit Nath (Author)

---

## Author Comment (AC4) · 14 Jul 2017

To The Editor, Subject: Manuscript Submitted (Revised version) online.

Dear respected Sir, Greetings Happy to submit our revised version today on 14 July 2017, as you suggested to improve. we have tried our best, to rectify our manuscript through by adopting theoretical model and redesign accordingly by following comments received from two reviewers, accordingly finished experimental results and revised the text.

Thanking you Sincerely yours Biswajit Nath (on behalf of all authors)